# Insights from a Pan India Sero-Epidemiological survey (Phenome-India Cohort) for SARS-CoV2

Salwa Naushin[1,2†], Viren Sardana[1,2†], Rajat Ujjainiya[1,2], Nitin Bhatheja[1], Rintu Kutum[1,2], Akash Kumar Bhaskar[1,2], Shalini Pradhan[1], Satyartha Prakash[1], Raju Khan[2,3], Birendra Singh Rawat[2,4], Karthik Bharadwaj Tallapaka[5], Mahesh Anumalla[5], Giriraj Ratan Chandak[2,5], Amit Lahiri[2,6], Susanta Kar[2,6], Shrikant Ramesh Mulay[2,6], Madhav Nilakanth Mugale[2,6], Mrigank Srivastava[2,6], Shaziya Khan[2,6], Anjali Srivastava[2,6], Bhawana Tomar[2,6], Murugan Veerapandian[2,7], Ganesh Venkatachalam[2,7], Selvamani Raja Vijayakumar[7], Ajay Agarwal[2,8], Dinesh Gupta[8], Prakash M Halami[2,9], Muthukumar Serva Peddha[2,9], Gopinath M Sundaram[2,9], Ravindra P Veeranna[2,9], Anirban Pal[2,10], Vinay Kumar Agarwal[10], Anil Ku Maurya[10], Ranvijay Kumar Singh[2,11], Ashok Kumar Raman[11], Suresh Kumar Anandasadagopan[2,12], Parimala Karuppanan[12], Subramanian Venkatesan[2,12], Harish Kumar Sardana[13], Anamika Kothari[13], Rishabh Jain[2,13], Anupama Thakur[2,13], Devendra Singh Parihar[2,13], Anas Saifi[2,13], Jasleen Kaur[2,13], Virendra Kumar[13], Avinash Mishra[2,14], Iranna Gogeri[2,15], Geethavani Rayasam[2,16], Praveen Singh[1,2], Rahul Chakraborty[1,2], Gaura Chaturvedi[1,2], Pinreddy Karunakar[1,2], Rohit Yadav[1,2], Sunanda Singhmar[1], Dayanidhi Singh[1,2], Sharmistha Sarkar[1,2], Purbasha Bhattacharya[1,2], Sundaram Acharya[1,2], Vandana Singh[1,2], Shweta Verma[1,2], Drishti Soni[1,2], Surabhi Seth[1,2], Sakshi Vashisht[1,2], Sarita Thakran[1,2], Firdaus Fatima[1,2], Akash Pratap Singh[1,2], Akanksha Sharma[1,2], Babita Sharma[1,2], Manikandan Subramanian[1], Yogendra S Padwad[2,17], Vipin Hallan[2,17], Vikram Patial[2,17], Damanpreet Singh[2,17], Narendra Vijay Tripude[2,17], Partha Chakrabarti[2,18], Sujay Krishna Maity[18], Dipyaman Ganguly[2,18], Jit Sarkar[2,18], Sistla Ramakrishna[2,19], Balthu Narender Kumar[19], Kiran A Kumar[19], Sumit G Gandhi[2,20], Piyush Singh Jamwal[20], Rekha Chouhan[20], Vijay Lakshmi Jamwal[2,20], Nitika Kapoor[2,20], Debashish Ghosh[2,21], Ghanshyam Thakkar[21], Umakanta Subudhi[2,22], Pradip Sen[2,23], Saumya Ray Chaudhury[2,23], Rashmi Kumar[2,23], Pawan Gupta[2,23], Amit Tuli[2,23], Deepak Sharma[2,23], Rajesh P Ringe[23], Amarnarayan D[24], Mahesh Kulkarni[2,25], Dhansekaran Shanmugam[2,25], Mahesh S Dharne[2,25], Sayed G Dastager[2,25], Rakesh Joshi[2,25], Amita P Patil[25], Sachin N Mahajan[25], Abujunaid Habib Khan[2,25], Vasudev Wagh[2,25], Rakesh Kumar Yadav[2,25], Ajinkya Khilari[2,25], Mayuri Bhadange[2,25], Arvindkumar H Chaurasiya[2,25], Shabda E Kulsange[2,25], Krishna Khairnar[2,26], Shilpa Paranjape[26], Jatin Kalita[27], Narahari G Sastry[27], Tridip Phukan[27], Prasenjit Manna[27], Wahengbam Romi[27], Pankaj Bharali[27], Dibyajyoti Ozah[27], Ravi Kumar Sahu[2,27], Elapavalooru VSSK Babu[2,28], Rajeev Sukumaran[2,29], Aiswarya R Nair[29], Prajeesh Kooloth Valappil[2,29], Anoop Puthiyamadam[2,29], Adarsh Velayudhanpillai[29], Kalpana Chodankar[2,30], Samir Damare[2,30], Yennapu Madhavi[2,31], Ved Varun Aggarwal[2,32], Sumit Dahiya[2,32], Anurag Agrawal[1,2], Debasis Dash[1,2*], Shantanu Sengupta[1,2*]

*For correspondence:
ddash@igib.res.in (DD);
shantanus@igib.res.in (SS)

†These authors contributed equally to this work

Competing interests: The authors declare that no competing interests exist.

[1]CSIR-Institute of Genomics and Integrative Biology, New Delhi, India; [2]Academy of Scientific and Innovative Research (AcSIR), Ghaziabad, India; [3]CSIR-Advanced Materials and Processes Research Institute, Bhopal, India; [4]CSIR-Central Building Research Institute, Roorkee, India; [5]CSIR-Centre for Cellular Molecular Biology, Hyderabad, India; [6]CSIR-Central Drug Research Institute, Lucknow, India; [7]CSIR-Central Electrochemical Research Institute, Karaikudi, India; [8]CSIR-Central Electronics Engineering Research Institute, Pilani, India; [9]CSIR-Central Food Technological Research Institute, Mysore, India; [10]CSIR-Central Institute of Medicinal Aromatic Plants, Lucknow, India; [11]CSIR-Central Institute of Mining and Fuel Research, Dhanbad, India; [12]CSIR-Central Leather Research Institute, Chennai, India; [13]CSIR-Central Scientific Instruments Organization, Chandigarh, India; [14]CSIR-Central Salt Marine Chemicals Research Institute, Bhavnagar, India; [15]CSIR Fourth Paradigm Institute, Bengaluru, India; [16]CSIR- Headquarters, Rafi Marg, New Delhi, India; [17]CSIR-Institute of Himalayan Bioresource Technology, Palampur, India; [18]CSIR-Indian Institute of Chemical Biology, Kolkata, India; [19]CSIR-Indian Institute of Chemical Technology, Hyderabad, India; [20]CSIR-Indian Institute of Integrative Medicine, Jammu, India; [21]CSIR-Indian Institute of Petroleum, Dehradun, India; [22]CSIR-Institute of Minerals and Materials Technology, Bhubaneswar, India; [23]CSIR-Institute of Microbial Technology, Chandigarh, India; [24]CSIR- National Aerospace Laboratories, Bengaluru, India; [25]CSIR-National Chemical Laboratory, Pune, India; [26]CSIR-National Environmental Engineering Research Institute, Nagpur, India; [27]CSIR-North - East Institute of Science and Technology, Jorhat, India; [28]CSIR-National Geophysical Research Institute, Hyderabad, India; [29]CSIR-National Institute for Interdisciplinary Science and Technology, Thiruvananthapuram, India; [30]CSIR-National Institute of Oceanography, Goa, India; [31]CSIR-National Institute of Science, Technology and Development Studies, New Delhi, India; [32]CSIR-National Physical Laboratory, New Delhi, India

**Abstract** To understand the spread of SARS-CoV2, in August and September 2020, the Council of Scientific and Industrial Research (India) conducted a serosurvey across its constituent laboratories and centers across India. Of 10,427 volunteers, 1058 (10.14%) tested positive for SARS-CoV2 anti-nucleocapsid (anti-NC) antibodies, 95% of which had surrogate neutralization activity. Three-fourth of these recalled no symptoms. Repeat serology tests at 3 (n = 607) and 6 (n = 175) months showed stable anti-NC antibodies but declining neutralization activity. Local seropositivity was higher in densely populated cities and was inversely correlated with a 30-day change in regional test positivity rates (TPRs). Regional seropositivity above 10% was associated with declining TPR. Personal factors associated with higher odds of seropositivity were high-exposure work (odds ratio, 95% confidence interval, p value: 2.23, 1.92–2.59, <0.0001), use of public transport (1.79, 1.43–2.24, <0.0001), not smoking (1.52, 1.16–1.99, 0.0257), non-vegetarian diet (1.67, 1.41–1.99, <0.0001), and B blood group (1.36, 1.15–1.61, 0.001).

## Introduction

The World Health Organization declared SARS-CoV-2 infection as a pandemic on March 11, 2020 (*WHO, 2020*). Within 2 weeks, India announced a lockdown strategy that severely influenced the growth of the pandemic, which was initially very focal in the large cities, gathering pace and spreading to smaller cities and towns as the nation unlocked for societal and economic considerations.

Early literature pointed towards asymptomatic transmission of SARS-CoV-2 and raised the need for extended testing (*Nishiura et al., 2020*; *Bai et al., 2020*). While RT-PCR was an undisputed choice for establishing a positive infection, serosurveillance revealed that many more were probably getting infected without manifesting symptoms (*Brown and Walensky, 2020*; *Oran and Topol,*

*2020*). Initial estimates of asymptomatic infection rate from the West were around 40–45% (*Oran and Topol, 2020*).

In India, the first case of Covid was reported on January 30, 2020 (*Andrews et al., 2020*). Serological surveys have confirmed that spread beyond the Indian megacities was minimal in early May–June, with less than 1% seropositivity outside the designated containment zones, suggesting that the lockdown had been effective in limiting the spread (*Murhekar et al., 2020a*). This was not without human and economic costs. By the end of June, migrant workforces caught in the cities during the lockdown were sent to their rural homes, which may have led to subsequent rapid, multifocal rise in cases in July 2020. By October 2020, total new cases began to decline with further outbreaks limited to a few geographies. Thus, the period of August–September 2020 is an important transition point. The present study is one of two studies at a national scale that was designed to assess the spread of infection. A national serosurvey in 70 districts of India, conducted by the Indian Council of Medical Research (ICMR), had a reported crude positivity rate of about 10% (*Murhekar et al., 2020b*). Since only a few districts of India account for the majority of urban areas, the ICMR survey is not representative of Indian cities (*Murhekar et al., 2020a*), where a combination of higher population density and indoor lives has led to greater spread of disease than rural areas. Existing city-wise serosurveys are variable in target population as well as methodology and cannot be easily compared to determine the relative course of the pandemic (*Ray et al., 2020*; *Satpati et al., 2020*; *Babu et al., 2020*; *Sharma et al., 2020*). The current study was launched by the Council of Scientific and Industrial Research (CSIR) in its more than 40 constituent laboratories and centers spread all over the country, representing a wide range of ethnicities, geosocial habitats, and occupational exposures in the form of a longitudinal cohort (Phenome-India Cohort). Though limited by not being a population-denominated study, the cohort includes permanent staff, families of staff members, students, and temporary employees proving support services such as security, sanitation,

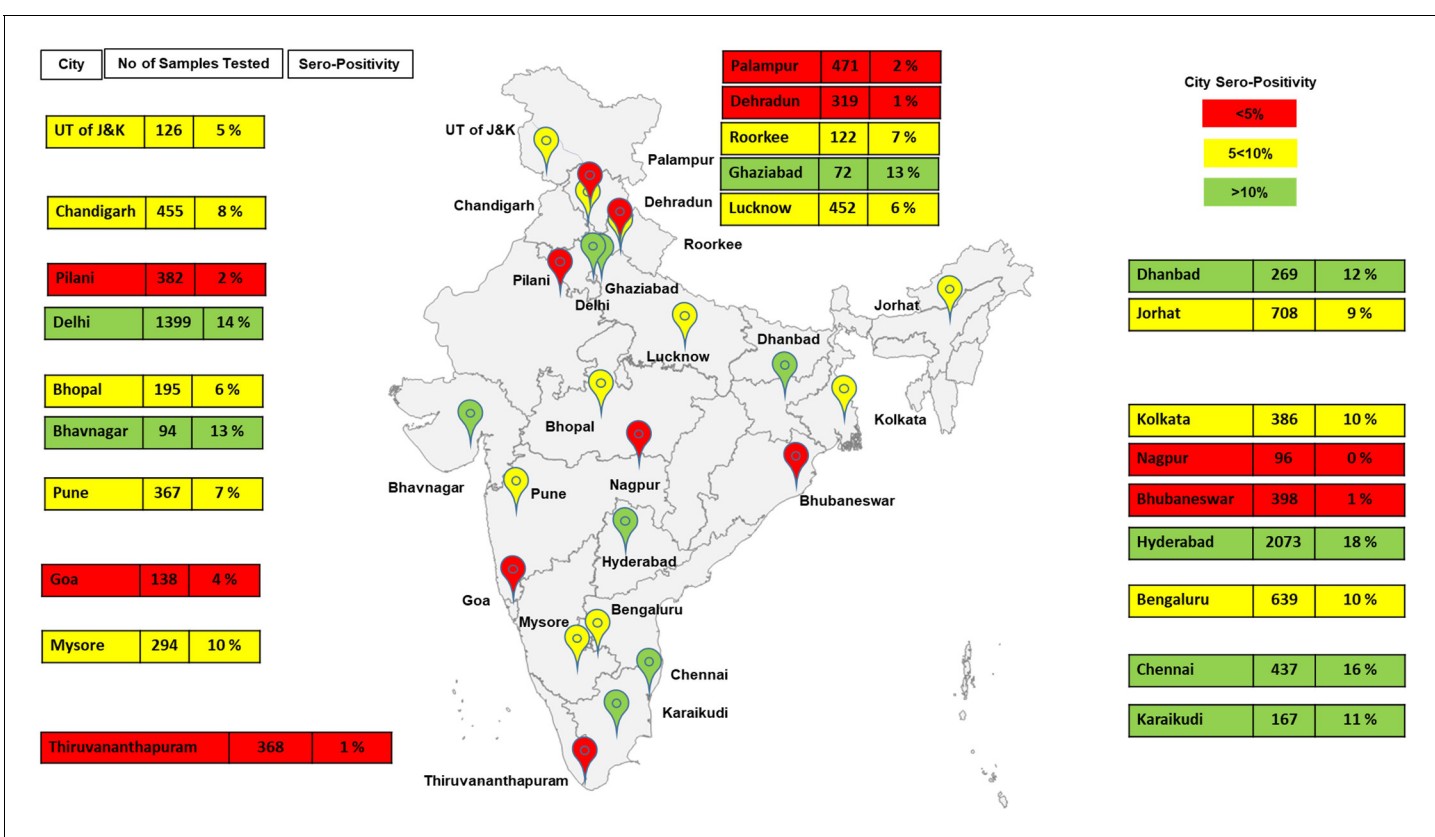

**Figure 1.** Demographics of serosurvey (India map may not be to scale and is for representation purposes only and seropositivity is rounded off). The online version of this article includes the following source data for figure 1:

**Source data 1.** Data for all labs/centers utilized for *Figure 2A,B*.

housekeeping, etc. This is a diverse microcosm of India encompassing multiple socioeconomic groups and has the advantage of permitting deeper assessments such as questionnaire surveys and periodic reassessment of humoral antibody response in those found to be seropositive. Our data is thus the first that permits valid comparisons between many important urban regions of India. Here, we report results from phase 1 of this study covering the critical period of August–September 2020.

## Results

### Seropositivity varied widely across India

In 10,427 subjects from over 17 states and 2 union territories, the average seropositivity was 10.14% (95% confidence interval [CI] 9.6–10.7), but varied widely across locations (*Figure 1*). We found that 95% of the seropositive subjects also had significant neutralizing activity, suggesting at least partial immunity (*Figure 1—source data 1*).

### Seropositivity, population density, and trajectory of new infections

As expected from the known outward spread of infection from large Indian cities, seropositivity was greater in regions with higher population density (*Figure 2A*). Changes in test positivity rate (TPR) are a relatively robust marker of the local level of transmission and are preferred when absolute number of tests or test rates are variable, as was the case here. Lab-wise seropositivity was correlated with the regional change in TPR. By this measure, regional transmission of SARS-CoV2 was inversely correlated to local seropositivity (*Figure 2B*). Seropositivity of 10% or more was associated with reductions in TPR, which may mean declining transmission (*Figure 2—source data 1*).

### Survey-based correlates of seropositivity

Out of 861 seropositive subjects who also provided data on symptomatology, 647 subjects (75.3%) did not recall any of the nine symptoms asked for (two of these did not provide gender data) (*Supplementary file 1*). Among the minority of subjects with symptoms, the most reported symptom constellation was those of a mild flu-like disease with fever (~50%) as the most frequent

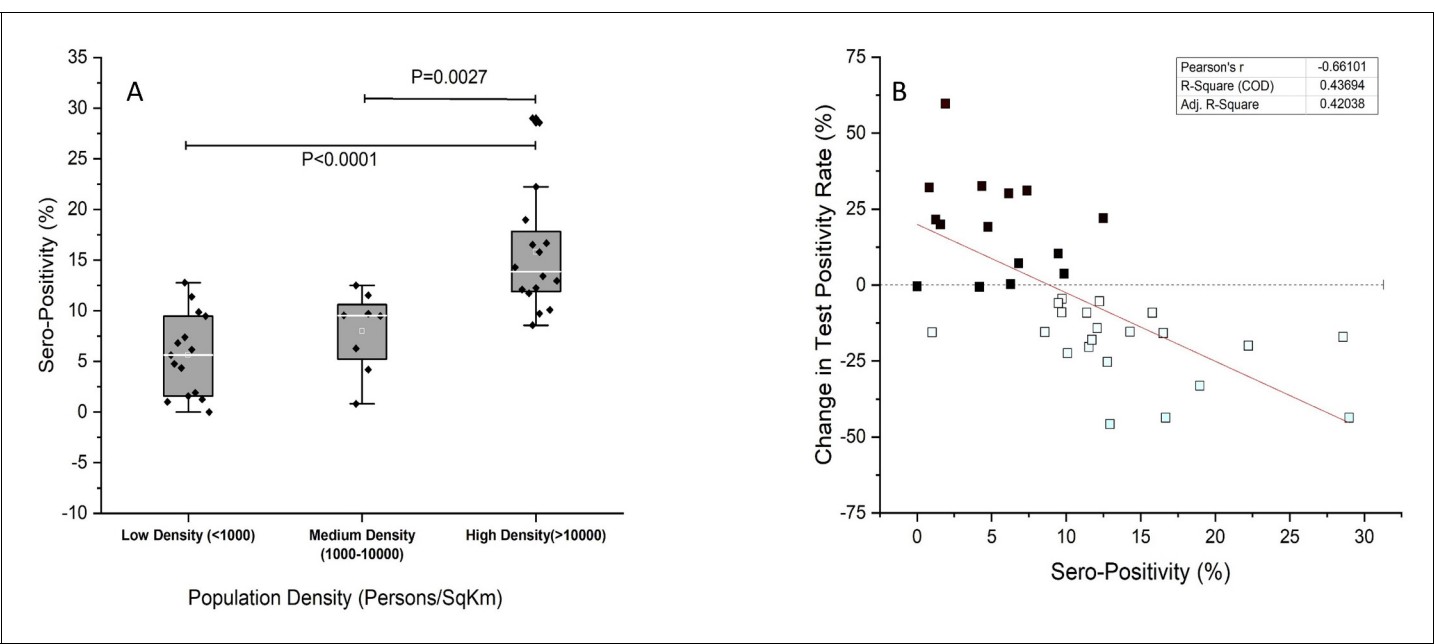

**Figure 2.** Seropositivity and test positivity rate. (**A**) Population density-based seropositivity, an overall p value of <0.0001 was obtained on one-way ANOVA. (**B**) Change in COVID19 test positivity rate (%) for states (y-axis) against observed seropositivity of labs/centers in the state (x-axis). A negative slope reflects declining test positivity rate with increase in seropositivity.

The online version of this article includes the following source data for figure 2:

**Source data 1.** Data for all labs/centers utilized for *Figure 2A, B*.

symptom. Loss of smell or taste was uncommonly reported (~25% of symptomatic subjects) (*Supplementary file 2*).

We further examined associations of other available variables with seropositivity to explore potential factors that modulate risk of infection in India. Apart from gender and age, distribution of the other variables recorded in CSIR-cohort (prevalence of smoking, diet, physiological parameters like ABO blood group type) was similar to the national averages and the sample can be considered representative (*Patidar and Dhiman, 2021*; *Mohan et al., 2018*; *Government of India, 2014*). The univariate associations are shown in *Figure 3*, separately for each gender. Due to gender imbalance and possible confounding between various parameters, significance of associations was further tested in a balanced iterative logistic regression (*Figure 3—figure supplement 1*). The strongest gender-independent associations were with occupation and mode of transport. Outsourced staff performing support services such as security, housekeeping, etc., and subjects using public transport were more likely to be seropositive. In males, smoking and vegetarian diet was associated with lower seropositivity.

Blood group type was reported for 7496 subjects. Blood group distribution of subjects in our study was similar to national reference based on a recent systematic review (*Patidar and Dhiman, 2021*). Seropositivity was significantly different between different groups, being highest for blood group type AB (10.19%) followed by B (9.94%), O (7.09%), and A (6.97%). Blood group O was found to be associated with a lower seropositivity rate, with an odds ratio (OR) of 0.76 (95% CI 0.64–0.91, p=0.018) vs. non-O blood group types, while B appeared to be at high risk with an OR of 1.36 (95% CI 1.15–1.61, p=0•001). While blood group A had an OR of 0.78, the association was not found to be significant (p=0.10) and a similar observation was made with blood group AB (p=0.35), it had an OR of 1.27 (*Supplementary file 3*). Rh factor was not found to have a significant association with seropositivity (p=0.35).

| Male | | | | Female | | |
|---|---|---|---|---|---|---|
| n | Odds Ratio (95% CI) | P-Value | | n | Odds Ratio (95% CI) | P-Value |
| **Occupation: Odds of Being Sero-Positive for Outsourced Staff when compared to Regular Staff** | | | | | | |
| 6457 | 2.05 (1.73-2.42) | <0.0001 | | 2487 | 2.75 (1.95-3.89) | <0.0001 |
| **Mode of Transport: Odds of Being Sero-Positive for Public Transport Users when compared to Private Transport Users** | | | | | | |
| 6375 | 1.91 (1.44-2.55) | <0.0001 | | 2398 | 1.83 (1.26-2.69) | 0.01 |
| **Diet Type: Odds of Being Sero-Positive for Non-Vegetarian Subjects when compared to Vegetarian Subjects** | | | | | | |
| 6345 | 1.78 (1.45-2.19) | <0.0001 | | 2382 | 1.33 (0.97-1.82) | 0.51 |
| **Smoking: Odds of Being Sero-Positive for Non-Smoking Subjects when compared to participants who Smoke** | | | | | | |
| 6379 | 1.62 (1.23-2.14) | 0.0058 | | | Data Not Sufficient | |

**Figure 3.** Demographics of data available for different variables.

The online version of this article includes the following source data and figure supplement(s) for figure 3:

**Source data 1.** Raw data for *Figure 3*.
**Source data 2.** Source file for tables show in *Figure 3*.
**Figure supplement 1.** Regression model of seropositivity.

## Stability of humoral response to SARS-CoV-2

Of 607 subjects whose samples were collected again at 3 months, anti-nucleocapsid (anti-NC) antibody levels were similar or higher for most, with 17 (2.8%) becoming seronegative (*Figure 4A*). In contrast, 34 subjects (5.6%) did not have neutralizing activity based on a surrogate measure (>20% inhibition of receptor-spike protein binding; *Figure 4B*). Of 175 subjects whose

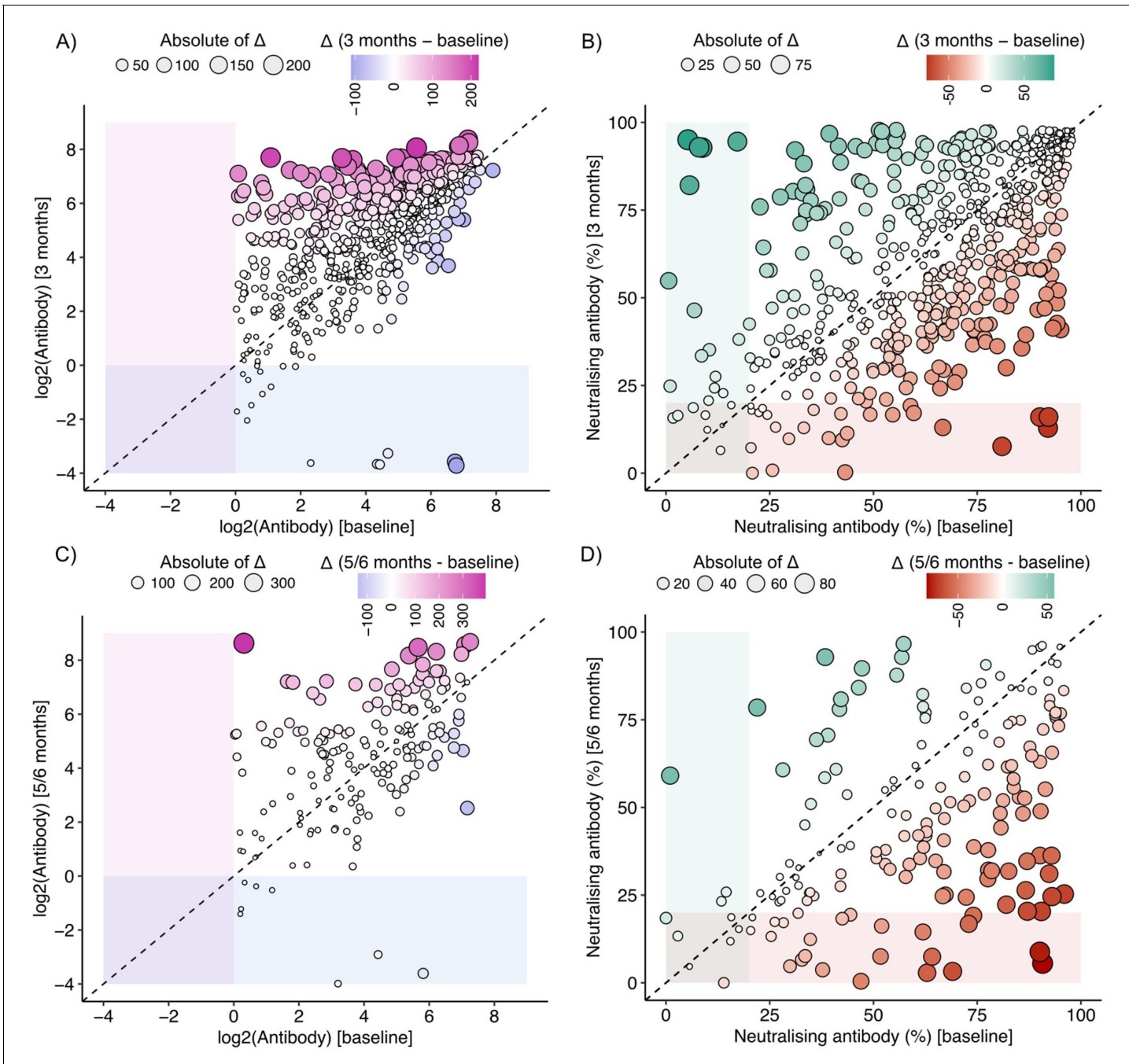

**Figure 4.** Antibody levels (A) and neutralizing activity (B) level at baseline (x-axis) and after 3 months (y-axis). Antibody levels (C) and neutralizing activity (D) level at baseline (x-axis) and after 5–6 months (y-axis).

The online version of this article includes the following source data for figure 4:

**Source data 1.** Raw data for antibody and neutralizing antibody levels at baseline, 3 months and 5-6 months.

samples were collected again at 5–6 months, 8 (4.6%) became seronegative (*Figure 4C*). In contrast, neutralizing activity was not present in 31 (17.7%) subjects (*Figure 4D*, *Figure 4—source data 1*).

## Discussion

This study, which recruited subjects from 24 cities in India, provides an important and timely snapshot of the spread of SARS-CoV2 pandemic across India shortly before the peak of new cases. It confirms that by September 2020 a large pool of recovered Indians with at least partial immunity existed. Between our study and the other national serosurvey at the same time, more than a hundred million Indians were likely to belong to this category. The subsequent nationwide decline of new cases starting in October 2020 can be well understood through these observations. There is some evidence of declining transmission in high-seropositivity regions within September, based on falling TPRs, but due to changes in tests and expansion of testing, this can only be indirectly inferred. As shown in our study, the fraction of such recovered subjects with resistance to reinfection was more than double among people performing high-contact jobs and using public transport. Thus, it is not surprising that in combination with a strong emphasis on masking and distancing, new cases started declining soon after this serosurvey. As of January 2021, despite the onset of winter and new year festivities, India has not seen major outbreaks. It is important to note that the crude seropositivity rate reported by us needs adjustments for demographics, fraction of infected subjects who may not develop antibodies, test characteristics, among others, to be a true population seropositivity rate. Here, we intentionally avoid such adjustments since it provides a false sense of precision with too many unknowns and unmeasurables. We focus on the more meaningful variation of the crude rate, its clinical correlates, and public health implications. We are confident from the data that a very large pool of recovered and immune subjects existed by September 2020, as stated, and expect that downward adjustments for national demographics will be counterbalanced by upward adjustments for almost 20% of RT-PCR-proven asymptomatic infections who develop transient antibody responses (*Koopmans and Haagmans, 2020*).

Apart from the seropositivity rate, our data also reveals important associations between demographic, physiological, lifestyle-related, and occupational attributes with susceptibility to infection. The workforce in our cohort comprised adult population, and no major difference was observed in seropositivity among different age groups or those with comorbidities. Males were found to be more susceptible, in agreement with other published reports (*Ortolan et al., 2020*). However, there were fewer females in our study and many of the occupational responsibilities with higher chances of exposure, like that of security personnel, were skewed towards males. On iteratively ran regression models, we did not find gender to be a predictor for seropositivity.

ABO blood group type has been shown to be associated with SARS-CoV-2 infection, but the results are variable in different studies. Most studies found that the O group is associated with a lower risk of infection or severity and blood group A was reported to be high risk in some studies (*Zhao et al., 2020*; *Wu et al., 2020*; *Göker et al., 2020*; *Ellinghaus et al., 2020*). In a meta-analysis authored by *Golinelli et al., 2020*, they observed positive association with A blood group, while blood group O was to be associated with lesser positivity using a random effects model. Another study from India observed blood group O to be associated with less mortality while blood group B with higher mortality when they analyzed the national data available (*Padhi et al., 2020*). A complex molecular interaction is said to play a significant role, and the molecular pathways for the same need to be elucidated for the effects observed especially with blood group O, which was also confirmed by our study.

The associations with diet and smoking are intriguing but preliminary. It has been proposed that a fiber-rich diet may play an important role in COVID-19 through anti-inflammatory properties by modification of gut microbiota (*Conte and Toraldo, 2020*). A recent review has highlighted the role of trace elements, nutraceuticals, and probiotics in COVID-19 (*Jayawardena et al., 2020*). The negative association with smoking has been reported elsewhere, but not shown to be causal (*Fontanet et al., 2020*; *Makoto Miyara et al., 2020*; *Rossato et al., 2020*; *Petrilli et al., 2020*; *Farsalinos et al., 2020*). Further exploration is necessary before reaching any conclusions, especially since seropositivity is an imperfect marker of infection risk and may equally well be explained by altered antibody response and dynamics.

The stability of antibody response is poorly understood, especially in India (*Figueiredo-Campos et al., 2020*; *Seow et al., 2020*; *Deshpande et al., 2020*). Our data shows that while anti-NC antibodies provide long-lasting evidence of viral exposure or infection, about 20% of seropositive individuals lack meaningful neutralization activity after 5–6 months. Using more stringent measures (more than 30% inhibition of surrogate receptor-spike protein binding), the loss of neutralization may be even higher. We speculate that this may be related to recurrence of outbreaks in March 2021, after the peak in September 2020.

The aggregate seropositivity of 10.14% in our multicentric study suggests that India had a large pool of recovered immune subjects by September 2020, especially among its high-contact workers and people using public transport, leading to a decline in new infections. However, the duration of such immunity may not be sufficient to prevent future outbreaks, even in highly affected regions.

## Materials and methods

### Study design, sampling, and data collection

The longitudinal cohort study was approved by the Institutional Human Ethics Committee of CSIR-IGIB vide approval CSIR-IGIB/IHEC/2019–20. The participation was voluntary, and participants had to fill an online informed consent with a consent to publish and share findings and deidentified data. Online statistical tool was utilized to calculate the minimum sample size for estimating seropositivity of about 5% with 10% precision (0.005) with 95% confidence to be 7300 (*Dhand and Khatkar, 2014*). In this study, we enrolled >10,000 subjects. 10,427 adult subjects working in the CSIR laboratories and their family members enrolled for the study based on voluntary participation. Informed consent was obtained from all the participants, and the samples were collected maintaining all recommended precautions. Blood samples (6 ml) were collected in EDTA vials from each participant, and antibodies to SARS-CoV-2 NC antigen were measured using an Electro-chemiluminescence Immunoassay (ECLIA)-Elecsys Anti-SARS-CoV-2 kit (Roche Diagnostics) as per the manufacturer's protocol. This approved assay is considered a method of choice when a single test is to be deployed (*Krüttgen et al., 2021*). A Cut-off index COI ≥1 was considered seropositive. Positive samples were further tested for neutralizing antibody (NAB) response directed against the spike protein using GENScript cPass SARS-CoV-2 Neutralization Antibody Detection Kit (GenScript, USA), according to the manufacturer's protocol. This is a blocking ELISA used for qualitative detection of total neutralizing antibodies against SARS-CoV-2 virus in plasma. A value of 20% or above was considered to have neutralizing ability.

All the participants were requested to fill an online questionnaire, which included information on date of birth, gender, blood group, type of occupation, history of diabetes, hypertension, cardiovascular disease, liver and kidney disease, diet preferences, mode of travel, contact history, and hospital visits. These forms were then downloaded in MS Excel data format and merged with registration forms filled at the time of sample collection based on unique IDs.

### Data and statistical analysis

Region-wise and total seropositivity was calculated from the fraction of samples positive for antibodies to SARS-CoV-2 NC antigen. Data regarding RT-PCR/rapid antigen testing and positive cases was gathered from http://www.covid19India.org. Change in TPR, a robust parameter for estimating the level of infection transmission when the level of testing is variable, was calculated as per the following equation:

$$Change\ in\ TPR = \frac{(TPR\ 15\ Days\ after\ DOC\ -\ TPR\ 15\ Days\ before\ DOC\ )}{Mean\ of\ TPR\ 15\ Days\ Prior\ and\ 15\ Days\ After} * 100$$

*DOC = date of collection.

IGIB, New Delhi, and NAL, Bengaluru, were removed from this change in TPR analysis for the sample collection was spread over 2–3 weeks in these labs. Questionnaire-based variables were assessed for response types and blank fields, that is, responses that were not provided by the participants of the survey. Based on multiple response types for each variable, categories were made to assign the response to either of the categories. For visualization, ggpubr (v0.4.0), ggrepel (v0.8.2), and ggplot2 (v3.3.2) packages were used in R. No data imputation was carried out. Chi-square test

was performed to evaluate variables that had a significant association with outcome of being tested positive ($p < 0.05$) along with OR with 95% CI. An adjusted p value was obtained through Bonferroni correction method for multiple comparison testing. Following the chi-square test, an iterative logistic regression was carried out on a balanced dataset. Variance inflation factor (VIF) was separately evaluated to assess multicollinearity. Statistical analysis and model development were carried out with visualization in R programming environment version 3.6.1, MS Excel 2016, and OriginPro V2021; faraway (v1.0.7) package was utilized for estimation of VIF.

### Role of the funding source

The sponsor of this study had no role in the study design, data collection, data analysis, data interpretation, or writing of the report. The corresponding authors had full access to all the data in the study and had final responsibility for the decision to submit for publication.

## Acknowledgements

SSG would like to acknowledge CSIR grant MLP 2007 for this work. SN, AKB, and RajatU would like to thank CSIR for their fellowships. SPrakash would like to acknowledge CSIR grant MLP-2002 (CSIR-IGIB) for the fellowship. NB would like to address CSIR grant GAP-0192 (CSIR-IGIB) for this work. We would also like to thank the directors of all the CSIR Institutes for facilitating the study. We also thank Pushpesh Ranjan, Jitendra, Neeraj Kumar, Abhijeet, and Rajkumar from AMPRI. V Santosh Kumar from CCMB. Dr. Chandra Prakash Pandey from CDRI. Vipul Sharma, Akansh Agarwal, Hansraj Choudhary, Vijay Chatterjee, Narendra Meena, Ved Prakash, Alok Mishra, Navin Singhal, Ankit Shukla, and Sudeep Rathore from CEERI. Avilash S, Rani C, and Naveen Shashidhar Kumbar from CFTRI. Swachchha Majumdar from CGCRI. Dr. Dyaneshwar Umrao Bawankule, Dr. Debabrata Chanda, Pankaj Shukla, Sanjay Singh, Dr. Dayanandan Mani, Ravi Kumar, Pankaj Yadav, and Parmanand Kumar from CIMAP. D C Sharma, Dr. Neelam J Gupta, A K Jain, and Sudhansu Bhagat from CRRI. Pankaj Pandey, Rajesh, Dr. Mohammed Faruq, and Ajay Pratap Singh from IGIB. Yogita Singh and Karvan Kaushal from CSIO. Jaykumar Patel, Shrikant Khandare, and Dr. Kannan Srinivasan from CSMCRI. Dr. Prakash L Ganapathi, G Bhatt, Shashikala U, and Shashidhar KN from NAL. Dr. Vidyadhar Mudkavi, Ravichandran C, and Sunil Babu MG from 4PI. Dr. Robin Singh, Mahesh S, Mohit Kumar Swarnakar, and Dr. Pankaj Kulurkar from IHBT. Saikat Chakrabarti and Sandip Paul from IICB. Siva Ranjith and B Vijay Kumar from IICT. Sajad Ahmed from IIIM. Rene Christina, Neha Bansal, and Ayan Banerjee from IIP. SK Goyal from NEERI-Delhi. Dr. PR Meganathan and Dr. Shaik Basha from NEERI-Hyderabad. Antara Sharma from NEIST. Dr. Shuchismita Benzwal and Chaitanya Dinesh from NGRI. Shana S Nair from NIIST. N Anandavalli and P Vasudevan from SERC. Vibha Malhotra Swaney from TKDL. Rashmi Arya and Prafulla Malwadkar from URDIP. Ajeet Singh and Dr. RK Sinha from HRDC.

## Additional information

### Funding

| Funder | Grant reference number | Author |
| --- | --- | --- |
| CSIR | MLP-2007 | Shantanu Sengupta |
| CSIR | MLP-2002 | Satyartha Prakash |
| Department of Science and Technology, India | GAP-0192 | Nitin Bhatheja |

The funders had no role in study design, data collection and interpretation, or the decision to submit the work for publication.

### Author contributions

Salwa Naushin, Resources, Data curation, Formal analysis, Supervision, Investigation, Writing - review and editing; Viren Sardana, Resources, Data curation, Software, Formal analysis, Supervision, Validation, Investigation, Visualization, Methodology, Writing - original draft, Writing - review and editing; Rajat Ujjainiya, Data curation, Formal analysis, Supervision, Validation, Investigation, Methodology,

Writing - review and editing; Nitin Bhatheja, Data curation, Software, Formal analysis, Validation, Investigation, Visualization, Methodology, Writing - review and editing; Rintu Kutum, Data curation, Software, Formal analysis, Validation, Visualization, Writing - review and editing; Akash Kumar Bhaskar, Resources, Formal analysis, Investigation, Methodology, Writing - review and editing; Shalini Pradhan, Formal analysis, Investigation, Methodology, Writing - review and editing; Satyartha Prakash, Data curation, Software, Formal analysis, Visualization, Writing - review and editing; Raju Khan, Birendra Singh Rawat, Karthik Bharadwaj Tallapaka, Mahesh Anumalla, Giriraj Ratan Chandak, Amit Lahiri, Susanta Kar, Shrikant Ramesh Mulay, Madhav Nilakanth Mugale, Mrigank Srivastava, Shaziya Khan, Anjali Srivastava, Bhawana Tomar, Murugan Veerapandian, Ganesh Venkatachalam, Selvamani Raja Vijayakumar, Ajay Agarwal, Dinesh Gupta, Prakash M Halami, Muthukumar Serva Peddha, Gopinath M Sundaram, Ravindra P Veeranna, Anirban Pal, Vinay Kumar Agarwal, Anil Ku Maurya, Ranvijay Kumar Singh, Ashok Kumar Raman, Suresh Kumar Anandasadagopan, Parimala Karuppanan, Subramanian Venkatesan, Harish Kumar Sardana, Anamika Kothari, Rishabh Jain, Anupama Thakur, Devendra Singh Parihar, Anas Saifi, Jasleen Kaur, Avinash Mishra, Iranna Gogeri, Geethavani Rayasam, Yogendra S Padwad, Vipin Hallan, Vikram Patial, Damanpreet Singh, Narendra Vijay Tripude, Partha Chakrabarti, Sujay Krishna Maity, Dipyaman Ganguly, Jit Sarkar, Sistla Ramakrishna, Balthu Narender Kumar, Kiran A Kumar, Sumit G Gandhi, Piyush Singh Jamwal, Rekha Chouhan, Vijay Lakshmi Jamwal, Nitika Kapoor, Debashish Ghosh, Ghanshyam Thakkar, Umakanta Subudhi, Pradip Sen, Saumya Ray Chaudhury, Rashmi Kumar, Pawan Gupta, Amit Tuli, Deepak Sharma, Rajesh P Ringe, Amarnarayan D, Mahesh Kulkarni, Dhansekaran Shanmugam, Mahesh S Dharne, Sayed G Dastager, Rakesh Joshi, Amita P Patil, Sachin N Mahajan, Abujunaid Habib Khan, Vasudev Wagh, Rakesh Kumar Yadav, Ajinkya Khilari, Mayuri Bhadange, Arvindkumar H Chaurasiya, Shabda E Kulsange, Krishna Khairnar, Shilpa Paranjape, Jatin Kalita, Narahari G Sastry, Tridip Phukan, Prasenjit Manna, Wahengbam Romi, Pankaj Bharali, Dibyajyoti Ozah, Ravi Kumar Sahu, Elapavalooru VSSK Babu, Rajeev Sukumaran, Aiswarya R Nair, Prajeesh Kooloth Valappil, Anoop Puthiyamadam, Adarsh Velayudhanpillai, Kalpana Chodankar, Samir Damare, Yennapu Madhavi, Ved Varun Aggarwal, Sumit Dahiya, Resources, Writing - review and editing, Contributed in lab/ centre co-ordination, enrolment of volunteers, sample collection and logistics; Virendra Kumar, Visualization, Writing - review and editing; Praveen Singh, Rahul Chakraborty, Gaura Chaturvedi, Pinreddy Karunakar, Rohit Yadav, Sunanda Singhmar, Dayanidhi Singh, Sharmistha Sarkar, Purbasha Bhattacharya, Sundaram Acharya, Vandana Singh, Shweta Verma, Drishti Soni, Surabhi Seth, Sakshi Vashisht, Sarita Thakran, Firdaus Fatima, Akash Pratap Singh, Akanksha Sharma, Babita Sharma, Resources, Formal analysis, Investigation, Writing - review and editing; Manikandan Subramanian, Formal analysis, Writing - review and editing; Anurag Agrawal, Conceptualization, Resources, Data curation, Formal analysis, Supervision, Funding acquisition, Validation, Investigation, Visualization, Writing - original draft, Project administration, Writing - review and editing; Debasis Dash, Conceptualization, Resources, Data curation, Software, Formal analysis, Supervision, Funding acquisition, Validation, Investigation, Visualization, Writing - original draft, Project administration, Writing - review and editing; Shantanu Sengupta, Conceptualization, Resources, Data curation, Formal analysis, Supervision, Funding acquisition, Validation, Investigation, Visualization, Methodology, Writing - original draft, Project administration, Writing - review and editing

## Author ORCIDs

Viren Sardana ![ORCID] https://orcid.org/0000-0002-6735-2946
Yogendra S Padwad ![ORCID] https://orcid.org/0000-0003-1793-9340
Partha Chakrabarti ![ORCID] https://orcid.org/0000-0001-9502-8695
Dipyaman Ganguly ![ORCID] https://orcid.org/0000-0002-7786-1795
Piyush Singh Jamwal ![ORCID] https://orcid.org/0000-0002-6640-7079
Deepak Sharma ![ORCID] http://orcid.org/0000-0003-1104-575X
Mahesh Kulkarni ![ORCID] http://orcid.org/0000-0003-3932-9092
Dhansekaran Shanmugam ![ORCID] http://orcid.org/0000-0002-3341-4846
Mahesh S Dharne ![ORCID] http://orcid.org/0000-0002-3965-7320
Ajinkya Khilari ![ORCID] http://orcid.org/0000-0002-5884-8747
Debasis Dash ![ORCID] https://orcid.org/0000-0002-5647-3785
Shantanu Sengupta ![ORCID] https://orcid.org/0000-0001-8461-0735

## Ethics

Human subjects: The longitudinal cohort study was approved by Institutional Human Ethics Committee of CSIR-IGIB vide approval CSIR-IGIB/IHEC/2019-20. The participation was voluntary, and participants had to fill an online informed consent with a consent to publish and share findings and deidentified data.

## Decision letter and Author response

Decision letter https://doi.org/10.7554/eLife.66537.sa1
Author response https://doi.org/10.7554/eLife.66537.sa2

## Additional files

### Supplementary files

• Supplementary file 1. Male:female distribution among asymptomatic and symptomatic seropositive individuals.

• Supplementary file 2. Frequency of symptoms in symptomatic seropositive individuals.

• Supplementary file 3. Distribution for blood group type in our cohort against national average and positivity among different blood groups. OR: odds ratio.

• Transparent reporting form

### Data availability

Source data files have been provided for Figures 1, 2,3 and 4. For Figure 2B, data is available in source data file and the district wise data listed in the source file on the number of confirmed cases and tests was taken from https://www.covid19india.org/. From this site, the data has to be manually extracted for desired time points by manually hovering over the number of cases and tests done graph when a district is selected after selection of the state. Code of Data Analysis and Model Definition Files- is available on Github at https://github.com/rintukutum/pan-india-csir-sero-epi (copy archived at https://archive.softwareheritage.org/swh:1:rev:7b789acbdb09c6842ab208c0ea84b52e88be43d7).

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
