## [Decision Letter]

**Acceptance summary:**

This publication is the first large scale COVID-19 seroprevalence study from India, and demonstrates that more than a hundred million individuals were infected. While this rate is notably higher than officially reported statistics, interestingly 3 month and 5-6 month follow-up analyses demonstrated that neutralization antibody activity significantly declines in about a quarter of individuals, most of whom remain seropositive. Overall this study’s findings have implications for reinfection as well as shed light on the severity of the initial COVID-19 infection wave in India.

**Decision letter after peer review:**

Thank you for submitting your article "Insights from a Pan India Sero-Epidemiological survey (Phenome-India Cohort) for SARS-CoV2" for consideration by *eLife*. Your article has been reviewed by 3 peer reviewers, and the evaluation has been overseen by a Reviewing Editor and Mone Zaidi as the Senior Editor. The following individuals involved in review of your submission have agreed to reveal their identity: Madhuri Kanitkar (Reviewer #2); Manindra Agrawal (Reviewer #3).

Essential revisions:

There was a marked debate amongst the reviewers regarding the validity of the findings and their usefulness. Overall, the reviewers felt that further expansion of the observational cohort make strengthen novel findings. There a numerous suggestions for improvement and narrowing focus on what can be concluded from the investigation.

1. Include three month follow-up data if available for the cohorts.

2. Narrow focus to limit conclusions on diet, smoking etc. as the reviewers felt that the study was not designed or powered to look at those differences.

*Reviewer #1 (Recommendations for the authors):*

My recommendation would be that the manuscript, while interesting, largely consists of a series of observations on a cohort that can hardly be said to be representative of the larger Indian population. It is this that I see to be the main defect of the manuscript. I tried hard to see if this study might at all be relevant to a broader understanding of COVID-19 in India but concluded that there were too many confounding variables – the selection of employees and contract workers in CSIR laboratories already speakers to a higher-than-average awareness of COVID-19. In addition, the bias towards metropolitan areas is a defect. The fact that the seroprevalence for the city of Pune provides results that are quite different from others for the same city is a cause for worry. As a last point, I thought that the writing could have been tighter and more targeted, especially in the introduction.

*Reviewer #2 (Recommendations for the authors):*

The authors of this well-designed cohort study for sero-positivity need to be complimented. A few suggestions for the study are as follows:

1. Table 1 – Title may specify Demographics of the Seropositive individuals.

2. The symptoms sought for in the history have not been clarified in the questionnaire and being recall minor symptoms are likely to be under reported. A table/figure may be added

3. It is not clear how some individuals have completed six months for repeat antibody titers when the study implies it was conducted in Aug Sept.

4. In case children of this close-knit cohort can be included the study can give an additional insight into the role children will play as schools open up.

*Reviewer #3 (Recommendations for the authors):*

No specific comments.

---

## [Author Response]

Essential revisions:There was a marked debate amongst the reviewers regarding the validity of the findings and their usefulness. Overall, the reviewers felt that further expansion of the observational cohort make strengthen novel findings. There a numerous suggestions for improvement and narrowing focus on what can be concluded from the investigation.1. Include three month follow-up data if available for the cohorts.

Thank you for the suggestion, which has led to important insights. While full cohort follow-up is not yet available, we have now obtained 3 month follow-up of 607 people who were seropositive at baseline (see revised Figure 4A and 4B), and 5-6 month follow-up for 175 individuals (now added as Figure 4C and 4D) a. We note that amongst those who passed a neutralization antibody surrogate threshold at baseline, 20-30% fail to pass that mark by 6 months. (Figure 4 Source Data) Yet, only 5% lose seropositivity, suggesting declining immunity despite persisting seropositivity. This may be relevant to the current course of increasing outbreaks after the previous peak in October 2020, and the manuscript discussion has been modified accordingly.

2. Narrow focus to limit conclusions on diet, smoking etc. as the reviewers felt that the study was not designed or powered to look at those differences.

We agree that the observations in regard to diet and smoking are only hypothesis generating and need specifically designed studies to confirm the findings. We have also mentioned in the manuscript that associations found between sero-positivity and some of the parameters should be confirmed with studies specifically designed for this purpose. As suggested, we state that “The associations with diet and smoking are intriguing, but preliminary. It has been proposed that a fiber-rich diet may play an important role in COVID-19 through anti-inflammatory properties by modification of gut microbiota.^25^ A recent review has highlighted the role of trace elements, nutraceuticals and probiotics in COVID-19.^26^ The negative association with smoking has been reported elsewhere, but not shown to be causal.^27-31^ Further exploration is necessary before reaching any conclusions, especially since seropositivity is an imperfect marker of infection-risk and may equally well be explained by altered antibody response and dynamics”

Reviewer #1 (Recommendations for the authors):My recommendation would be that the manuscript, while interesting, largely consists of a series of observations on a cohort that can hardly be said to be representative of the larger Indian population. It is this that I see to be the main defect of the manuscript. I tried hard to see if this study might at all be relevant to a broader understanding of COVID-19 in India but concluded that there were too many confounding variables – the selection of employees and contract workers in CSIR laboratories already speakers to a higher-than-average awareness of COVID-19. In addition, the bias towards metropolitan areas is a defect. The fact that the seroprevalence for the city of Pune provides results that are quite different from others for the same city is a cause for worry. As a last point, I thought that the writing could have been tighter and more targeted, especially in the introduction.

We agree with the reviewer that this is a very specific cohort, largely urban, and with higher levels of education than average. We further agree that the utility of this cohort is not in making general statements about the population, but rather in deriving specific insights for which the cohort is best suited. We enumerate some of them that are present in this manuscript.

a. It is as important to understand the relative degree of spread between Indian cities, where a combination of denser population and indoor lives has led to the greatest spread of disease. Since pandemics are typically self-limiting, regions with greater spread are further along the course and can expect declines faster. This provides useful insight for public health strategy. While our cohort does not necessarily represent the average population, it is similar between cities, something that is not true for any other survey. The ICMR national sero-survey is a random selection of districts and is heavily rural biased.^1^ While that is important, that is not where fast growing outbreaks are likely based on a very outdoor life and lower density. Other city-wise serosurveys are variable in target population as well as methodology and cannot be easily compared.^2-5^ Thus our data is the first that permits comparison between many important urban regions of India, showing which regions were more advanced along the course and where future outbreaks were still likely. We note here that some of the regions identified by this survey as high risk such as Kerala, interior Maharashtra, amongst others, are where the outbreaks continued until much later.

b. The CSIR cohort has the added advantage of greater baseline data and repeated access, we are able to determine antibody stability, as shown, and possible correlates

c. The cohort is well suited to understanding clinical associations of SARS CoV2 infections such as symptom rate and severity amongst its participants as well as associations of infection risks (using seropositivity as an imperfect surrogate).

d. The Pune city sero-surveillance which has been pointed out by the reviewer was a survey of Pune’s five most affected sub-wards and not the Pune population in general. ^6^ Despite all the limitations, which we accept in the prior comment, our overall crude positivity rate of 10% is very similar to that of the ICMR national serosurvey, and in general the patterns we see are along the lines of what is known about severity of outbreaks. Thus, there is no real evidence to the contrary that would establish inaccuracy of the trends seen by us, and we respectfully note that surprising findings may be the most valuable ones. In fact, seeing current trends of rising cases in Maharashtra, including in Pune, when compared to other cities, our survey values may have been more correct.

Reviewer #2 (Recommendations for the authors):The authors of this well-designed cohort study for sero-positivity need to be complimented. A few suggestions for the study are as follows:1. Table 1 – Title may specify Demographics of the Seropositive individuals.

Thank you for the correction. It has been addressed and the title now changed.

2. The symptoms sought for in the history have not been clarified in the questionnaire and being recall minor symptoms are likely to be under reported. A table/figure may be added

We regret the lack of clarity. Supplementary File 2 contains the frequency of symptoms reported as per the questionnaire. We agree that minor symptoms are usually underreported, but during these times of SARS-CoV-2 pandemic, people have become highly aware of their symptoms.

3. It is not clear how some individuals have completed six months for repeat antibody titers when the study implies it was conducted in Aug Sept.

For CSIR-IGIB, the study started in May-June and hence we were able to obtain minimal samples at 6 months completion. This is also the reason, CSIR-IGIB had been removed from figure 2B analysis as stated in the manuscript.

4. In case children of this close-knit cohort can be included the study can give an additional insight into the role children will play as schools open up.

We agree that children could provide specific insights into transmission dynamics specifically when schools open up but the current ethical approval didn’t permit us to have reporting for children and hence would be taken care of in future.

References:

1. Murhekar M, Bhatnagar T, Selvaraju S, et al. Prevalence of SARS-CoV-2 infection in India: Findings from the national serosurvey, May-June 2020. Indian Journal of Medical Research 2020;152(1):48-60. doi: 10.4103/ijmr.IJMR_3290_20

2. Ray A, Singh K, Chattopadhyay S, et al. Seroprevalence of anti-SARS-CoV-2 IgG antibodies in hospitalized patients at a tertiary referral center in North India. medRxiv 2020:2020.08.22.20179937. doi: 10.1101/2020.08.22.20179937

3. Satpati P, Sarangi SS, Gantait K, et al. Sero-surveillance (IgG) of SARS-CoV-2 among Asymptomatic General population of Paschim Medinipur District, West Bengal, India(Conducted during last week of July and 1st week of August 2020) – A Joint Venture of VRDL Lab (ICMR), Midnapore Medical College and amp; Hospital and amp; Department of Health and Family Welfare,Govt. of West Bengal, Paschim Medinipur. medRxiv 2020:2020.09.12.20193219. doi: 10.1101/2020.09.12.20193219

4. Babu GR, Sundaresan R, Athreya S, et al. The burden of active infection and anti-SARS-CoV-2 IgG antibodies in the general population: Results from a statewide survey in Karnataka, India. medRxiv 2020:2020.12.04.20243949. doi: 10.1101/2020.12.04.20243949

5. Sharma N, Sharma P, Basu S, et al. The seroprevalence and trends of SARS-CoV-2 in Delhi, India: A repeated population-based seroepidemiological study. medRxiv 2020:2020.12.13.20248123. doi: 10.1101/2020.12.13.20248123

6. Kulkarni P. “Pune’s first sero-survey shows 51.5% citizens have Covidantibodies”: The Times of India, 18 August 2020.; 2020 [Available from: https://timesofindia.indiatimes.com/city/pune/citys-first-sero-survey-shows-51-5-citizens-have-covid-antibodies/articleshow/77602008.cms].

7. Stadlbauer D, Tan J, Jiang K, et al. Repeated cross-sectional sero-monitoring of SARS-CoV-2 in New York City. Nature 2021;590(7844):146-50. doi: 10.1038/s41586-020-2912-6

8. Bauch CT. Estimating the COVID-19 *R* number: a bargain with the devil? The Lancet Infectious Diseases 2021;21(2):151-53. doi: 10.1016/S1473-3099(20)30840-9

9. Mandal M, Mandal S. COVID-19 pandemic scenario in India compared to China and rest of the world: a data driven and model analysis. medRxiv 2020:2020.04.20.20072744. doi: 10.1101/2020.04.20.20072744

10. 2014 [cited 2020]. [Available from: https://censusindia.gov.in/2011-Common/Sample_Registration_System.html].

11. Conte L, Toraldo DM. Targeting the gut–lung microbiota axis by means of a high-fibre diet and probiotics may have anti-inflammatory effects in COVID-19 infection. Therapeutic Advances in Respiratory Disease 2020;14:1753466620937170. doi: 10.1177/1753466620937170

12. Jayawardena R, Sooriyaarachchi P, Chourdakis M, et al. Enhancing immunity in viral infections, with special emphasis on COVID-19: A review. Diabetes and metabolic syndrome 2020;14(4):367-82. doi: 10.1016/j.*dsx*.2020.04.015 [published Online First: 2020/04/26]

13. Patidar GK, Dhiman Y. Distribution of ABO and Rh (D) Blood groups in India: A systematic review. ISBT Science Series;n/a(n/a) doi: https://doi.org/10.1111/voxs.12576

14. Golinelli D, Boetto E, Maietti E, et al. The association between ABO blood group and SARS-CoV-2 infection: A meta-analysis. PLOS ONE 2020;15(9):e0239508. doi: 10.1371/journal.pone.0239508

15. Zhao J, Yang Y, Huang H, et al. Relationship between the ABO Blood Group and the COVID-19 Susceptibility. medRxiv 2020:2020.03.11.20031096. doi: 10.1101/2020.03.11.20031096

16. Wu Y, Feng Z, Li P, et al. Relationship between ABO blood group distribution and clinical characteristics in patients with COVID-19. Clinica Chimica Acta 2020;509:220-23. doi: https://doi.org/10.1016/j.cca.2020.06.026

17. Latz CA, DeCarlo C, Boitano L, et al. Blood type and outcomes in patients with COVID-19. Ann Hematol 2020;99(9):2113-18. doi: 10.1007/s00277-020-04169-1 [published Online First: 07/12]

18. Göker H, Aladağ Karakulak E, Demiroğlu H, et al. The effects of blood group types on the risk of COVID-19 infection and its clinical outcome. Turk J Med Sci 2020;50(4):679-83. doi: 10.3906/sag-2005-395

19. Barnkob MB, Pottegård A, Støvring H, et al. Reduced prevalence of SARS-CoV-2 infection in ABO blood group O. Blood Adv 2020;4(20):4990-93. doi: 10.1182/bloodadvances.2020002657